# Digital Mental Health Interventions: Differences in Diet Culture Intervention Framing

**DOI:** 10.3390/ijerph21010024

**Published:** 2023-12-23

**Authors:** Hannah F. Fitterman-Harris, Gabrielle G. Davis, Samantha P. Bedard, Claire E. Cusack, Cheri A. Levinson

**Affiliations:** 1Department of Psychological and Brain Sciences, University of Louisville, Louisville, KY 40292, USA; sspoor@uwyo.edu (S.P.B.); claire.cusack@louisville.edu (C.E.C.); cheri.levinson@louisville.edu (C.A.L.); 2College of Education and Human Development, University of Louisville, Louisville, KY 40292, USA; gabrielle.davis@louisville.edu; 3Department of Psychology, University of Wyoming, Laramie, WY 82071, USA; 4Department of Pediatrics, Division of Child and Adolescent Psychiatry and Psychology, University of Louisville, Louisville, KY 40292, USA

**Keywords:** diet culture, wellness, digital mental health interventions, single-session interventions, eating disorders

## Abstract

Diet culture is a societal norm that ranks thin bodies as superior to other body types and has been associated with negative outcomes, such as eating disorders. Wellness has evolved into a term that is often used to promote diet culture messages. One possible way to combat diet culture is through single-session, digital mental health interventions (DMHIs), which allow for increased access to brief public health treatments. The framing of DMHIs is critical to ensure that the target population is reached. Participants (*N* = 397) were enrolled in a single-session DMHI, which was framed as either a *Diet Culture Intervention* (*n* = 201) or a *Wellness Resource* (*n* = 196). Baseline group differences in eating disorder pathology, body image, weight stigma concerns, fat acceptance, and demographic characteristics were analyzed. Across groups, participants reported moderately high eating disorder pathology, low-to-moderate levels of body dissatisfaction, moderate levels of fat acceptance, and either very low or very high weight stigma concerns. Participants in the *Diet Culture Intervention* group reported higher levels of fat acceptance than those in the *Wellness Resource* group (*p* < 0.001). No other framing group differences were identified, though post hoc analyses revealed differences based on recruitment source (i.e., social media versus undergraduate research portal). This study found that framing a DMHI as targeting diet culture or as a Wellness Resource can result in the successful recruitment of individuals at risk of disordered eating. Framing a DMHI as a *Wellness Resource* may increase recruitment of individuals with low levels of fat acceptance, which may be particularly important for dismantling diet culture, disordered eating, and weight stigma concerns. Future research should assess DMHI framing in other populations, such as men and adolescents.

## 1. Introduction

*Diet Culture.* Diet culture is a type of societal norm that classifies thin bodies as righteous and superior and large bodies as immoral and inferior [1]. Myths about food and health are also characteristic of diet culture, such as the false belief that an individual’s weight is synonymous with—and a direct representation of—their health [1]. Diet culture, along with other environmental factors (e.g., social support, familial context) [2], influences the development and maintenance of eating disorders (EDs). Targeting factors that exacerbate ED risk, such as diet culture, may be important in the prevention or treatment of EDs, thereby reducing the individual and societal burden. Decreasing the ED risk and burden is especially notable given that EDs have the second highest mortality rate of any psychiatric illnesses and are on the rise, with hospitalization rates doubling after the start of COVID-19 [3,4].

*Weight Stigma Concerns.* In addition to contributing to EDs, diet culture reinforces weight stigma [1]. Weight stigma includes negative attitudes, stereotypes, and discrimination that devalue individuals in larger bodies (e.g., believing that people in larger bodies are lazy) [5,6]. Experienced weight stigma and internalized weight stigma are associated with many adverse physical and psychological consequences, including chronic inflammation; elevated cortisol levels; and increased rates of anxiety, depression, disordered eating behaviors, body dissatisfaction, substance use disorders, and suicidality [7,8,9,10,11]. Further, concerns about the possibility of experiencing weight stigma (which are often fueled by past experiences of weight stigma) negatively influence physical and mental health [12]. For example, many individuals in larger bodies avoid or delay medical appointments due to fears of weight stigma [13].

*Digital Single-Session Interventions.* Despite these notable negative outcomes, accessible information, education, and intervention against diet culture remain sparse. Two ways to reach a large audience quickly to change beliefs are through (a) digital interventions and (b) single-session interventions [14,15,16,17]. Digital mental health interventions (DMHIs) are mental health services that are implemented using online or mobile formats to augment or serve in place of in-person treatments [14]. DMHIs enable greater access to treatment than in-person services [14] by removing or reducing barriers related to transportation, scheduling, and cost [15]. These interventions are critical, as most individuals with mental illness do not receive treatment, often due in part to these barriers [18]. Though some treatment barriers are resolved through the implementation of DMHIs, retention is generally low [19], with up to 50% attrition [20]. Single-session DMHIs attempt to address retention concerns through a reduced time commitment, while maintaining similar levels of efficacy as longer interventions [16,17].

*Framing of Digital Mental Health Interventions.* The framing of DMHIs (i.e., how DMHIs are described in recruitment materials) is particularly important, as different framings can result in differing levels of engagement [21,22]. Successful framing helps maximize recruitment [22], increasing the reach of DMHIs. Additionally, framing greatly influences who enrolls in the intervention and therefore contributes to whether the target population is reached [22]. For example, some individuals may be reluctant—or unaware of the need—to challenge diet culture messages but may be more receptive to a different framing. One study of military veterans found that nearly half of all participants were recruited to complete an online health survey through just 2 of 15 different framings that were posted on Facebook: one that emphasized in the headline an incentive for participation and one that highlighted sharing the post with a veteran, compared to those that highlighted other messages, such as “social norms” and “empowerment” [21]. Another study found that men were more likely to sign up for an online mental health study when Facebook advertisements centered “strength” in the framing, though notably these participants had the lowest levels of engagement on the study website [22]. These differences in participant recruitment showcase the importance of understanding which messages will be most appealing to the intended population.

Content related to wellness may seem like one way to counteract diet culture, as the term wellness is formally defined as being in “good health,” as defined as of “sound body, mind, or spirit” [23,24]; however, “wellness” media content has evolved into promoting diet culture messages, such as weight loss strategies [25]. Individuals who are at risk of ED pathology may in fact be more drawn to interventions or resources marketing “wellness,” rather than interventions framed as challenging diet culture due to the ego-syntonic nature of EDs (i.e., rigid beliefs about food, fatness, and diet culture are consistent with the individuals’ sense of self) [26]. However, it is unknown if there are differences in who will be recruited and who will engage with DMHIs based on the framing of the intervention (diet culture versus wellness). It is critical to reach individuals at scale due to the pervasiveness and harms of diet culture, and because diet culture is upheld through systems of oppression (e.g., anti-fat discrimination). To ensure DMHIs reach the target population, it is necessary to characterize who enrolls in the intervention, as this information may inform future recruitment strategies.

As such, the purpose of the current study is to determine whether two forms of framing an intervention aimed at dismantling diet culture attitudes were effective for recruitment of individuals with elevated ED pathology, negative body image, and weight stigma concerns, and low levels of fat acceptance (i.e., our target population). Specifically, this study sought to assess whether differences in the framing of the intervention would result in differences in participant characteristics between the framing groups. Because of the positive beliefs about dieting that individuals with increased ED pathology endorse [26], we hypothesized that individuals with higher levels of weight stigma concerns and higher levels of ideal muscularity would be more likely to enroll in the DMHI study when it was framed as a Wellness Resource compared to a Diet Culture Intervention. Additionally, we hypothesized that those who enroll in the Wellness Resource would have comparable or higher ED pathology compared to those who enrolled in the Diet Culture Intervention. We further hypothesized that participants who enrolled in the Wellness Resource would report more positive beliefs about restriction compared to participants who enrolled in the Diet Culture Intervention. Hypotheses were preregistered on Open Science Framework (osf.io/yv5ua) prior to any data analyses. Of note, two hypotheses were modified after preregistration. The measures used within this study assess beliefs about restriction, rather than actual restriction. Our hypothesis was adjusted accordingly. Further, the rate of beliefs about restriction in the general population is not established, preventing a direct comparison. We therefore modified our hypothesis to compare beliefs about restriction among participants that were enrolled in the two framings. Additionally, the preregistration discusses differences in weight stigma instead of weight stigma concerns, which is what was measured in this study.

## 2. Materials and Methods

### 2.1. Procedures

This study was approved by the Institutional Review Board of the University of Louisville. The current study was part of a larger single-session DMHI study testing the efficacy of the intervention in terms of decreasing ED pathology and weight stigma concerns, as well as improving body image and fat acceptance. Individuals were recruited through social media platforms (e.g., Instagram), the research laboratory’s website, and an undergraduate student research portal. In recruitment materials, the intervention was framed either as a Diet Culture Intervention or a Wellness Resource (see Figure 1). Individuals were eligible to sign up for the study through one of the two framings. Recruitment materials for both framings were posted on all recruitment outlets, though these framings were not advertised as being associated with the same study and individuals therefore were not guaranteed to see both framings. If an individual attempted to enroll in the study through both the Diet Culture Intervention and Wellness Resource framings, they were asked which one they would prefer to complete and provided the link to the selected framed intervention.

### 2.2. Participants

For both the larger study and the current study, participants needed to be at least 18 years old to enroll. This study and the larger study were specifically designed for women, inclusive of individuals who identified as women or who reported female sex assigned at birth and identified as nonbinary. Nonbinary individuals assigned female sex at birth were included in this study because they are likely to have been socialized in ways that are similar to women and therefore are likely to have been the target of diet culture messaging. Participants were excluded if they identified as men (*n* = 8). Additionally, participants were excluded if they failed the attention check item (*n* = 14). Duplicate completions were removed (*n* = 39). Across groups, 397 participants completed at least baseline demographic information and were included in analyses. When framed as a *Diet Culture Intervention*, 201 participants signed up, compared to 196 who enrolled when it was framed as a *Wellness Resource*. See Table 1 for additional demographic information.

### 2.3. Measures

#### 2.3.1. ED Pathology

The State Eating Disorder Symptom Survey [27] is a 30-item measure of state-based ED thoughts and urges (e.g., “I want to have an empty stomach”). This instrument was modeled after the EDE-Q-6, an established measure of ED pathology [28]. Items are rated on a scale from 0 to 6, with higher scores reflecting more eating disorder symptoms. Cronbach’s alpha for the current study was *α* = 0.96.

The Thinness and Restricting Expectancies Inventory [29] is a 44-item measure of expected reinforcement for dieting, thinness, and restriction (e.g., “I would be happy if I were thin”). For this study, the 14 items related specifically to beliefs about restriction (e.g., “When I limit what I eat, others respect me”) were also analyzed separately as a subscale to assess differences in beliefs about restriction. Items are rated on a scale from 1 to 7, with higher scores representing more positive expectancies related to thinness and restriction. Cronbach’s alpha was *α* = 0.98 for all items and *α* = 0.96 for the restriction items.

The Eating Disorder Fears Questionnaire [30] was included to assess fear of weight gain (e.g., “I fear gaining weight”). The fear of weight gain subscale contains two items rated on a scale from 1 to 7, with higher scores reflecting more fear of weight gain. Cronbach’s alpha was *α* = 0.96.

#### 2.3.2. Body Image

The Thin Ideal Questionnaire [31] consists of two subscales: Body Dissatisfaction and Body Ideal (e.g., “Slim women are more attractive”). A 17-item, modified version of this scale was used for the current study. Items are rated on a scale from 1 to 5, with higher scores representing greater body dissatisfaction and more belief in the “ideal body”. Cronbach’s alpha was *α* = 0.91 and *α* = 0.92 for the Body Dissatisfaction and Body Ideal subscales, respectively.

The Female Body Scale and Female Fit Body Scale [32] each consist of nine drawn figures with varying degrees of adiposity and muscularity, respectively. Body dissatisfaction is assessed by computing discrepancies between which figure participants rate that they *want* their body to look like and which their body *actually* looks like in terms of adiposity for the Female Body Scale and muscularity for the Female Fit Body Scale [32]. Scores can range from −8 to 8, with scores further from zero (positive or negative) indicating greater dissatisfaction. For this study, ideal muscularity was determined by the rating selected for the item about how they *wanted* their body to look on the Female Fit Body Scale, with higher scores reflecting higher levels of desired muscularity.

The Body Appreciation Scale-2 [33] was included as a 10-item measure of positive body image (e.g., “I respect my body”). Items are rated on a scale from 1 to 5, with higher scores reflecting more body appreciation. Cronbach’s alpha was *α* = 0.96.

#### 2.3.3. Weight Stigma Concerns

The Weight Stigma Concerns Scale [12] contains five items and measures worries about social negativity due to weight (e.g., “I am afraid that other people will reject me because of my weight”). Items are rated on a scale from 1 to 7, with higher scores indicating more concerns about weight stigma. Cronbach’s alpha was *α* = 0.96.

#### 2.3.4. Fat Acceptance

The Fat Acceptance Scale [34] contains 26 items and assesses levels of fat-accepting attitudes, beliefs, and behaviors (e.g., “Society should encourage more positive attitudes towards fat people”). Items are rated on a scale from 1 to 6, with higher scores representing more fat acceptance. Cronbach’s alpha was *α* = 0.91.

#### 2.3.5. Scoring Thresholds

Measures used in this study do not have specific clinical cut points. Scores were considered moderate if they were in the middle range of the rating scale (e.g., between 3 and 5 on a scale from 1 to 7). Scores were considered elevated if they were in the moderate range or higher.

### 2.4. Data Analyses

Statistical analyses were conducted using IBM SPSS Version 29. Mean scores were examined across groups. Chi-square tests, Fisher’s exact tests, and *t*-tests were conducted to assess differences between the framing groups. The Weight Stigma Concerns Scale scores were bimodal; therefore, a Wilcoxon rank-sum test was conducted. Given the number of tests, a Benjamini–Hochberg correction was implemented to account for the false discovery rate [35].

## 3. Results

### 3.1. Overall Descriptives

Across groups, participants reported moderately high ED pathology, low-to-moderate levels of body dissatisfaction, moderate levels of body appreciation, very low or very high weight stigma concerns, and moderate fat acceptance (See Table 2).

### 3.2. Differences between Groups

Participants in the *Diet Culture Intervention* group reported higher scores on the Fat Acceptance Scale (*p* < 0.001), representing higher levels of fat acceptance compared to individuals who enrolled in the study when it was framed as a *Wellness Resource* (see Table 3). While not meeting the Benjamini–Hochberg critical *p*-value, participants enrolled in the *Diet Culture Intervention* were younger and more participants identified as multiracial compared to participants enrolled in the *Wellness Resource*. Participants across groups reported comparable ED pathology on the Fear of Weight Gain subscale of the Eating Disorder Fears Questionnaire, State Eating Disorder Symptom Survey, Thinness and Restricting Expectancies Inventory, and restriction items of the Thinness and Restricting Expectancies Inventory (*p*s > 0.05). There were also no group differences for student status, sexual orientation, body mass index, the Body Dissatisfaction or Ideal Body subscales of the Thin Ideal Questionnaire, Female Body Scale, Female Fit Body Scale, the Ideal Muscularity item of the Female Fit Body Scale, Body Appreciation Scale, or the Weight Stigma Concerns Scale (*p*s > 0.05; see Table 3).

### 3.3. Differences between Recruitment Sources

Post hoc analyses were conducted to determine whether there were differences among participants based on where participants were recruited—from social media (inclusive of the research lab’s website) or from the undergraduate research portal. Several differences emerged, with higher levels of reported ED pathology, poorer body image, and more weight stigma concerns among participants recruited through social media (*p* < 0.01; See Table 4). Participants recruited through social media were also older, less likely to be a student, more likely to identify as White, and less likely to identify as Black compared to those recruited through the undergraduate research portal. Importantly, there was no difference between the framing groups regarding recruitment source, *ꭓ*^2^(1) = 3.03, *p* > 0.05.

## 4. Discussion

We examined whether individuals who enrolled in an intervention framed as a *Diet Culture Intervention* differed from those who enrolled in a *Wellness Resource*. These findings suggest that framing an intervention that is designed to dismantle diet culture as a *Wellness Resource* may increase recruitment of individuals with lower levels of fat acceptance compared to framing it as a *Diet Culture Intervention*. Fat acceptance may be particularly important for dismantling diet culture, addressing disordered eating, and reducing weight stigma concerns. These results are especially noteworthy for future interventions specifically targeting fat acceptance. Both framings of the intervention successfully recruited individuals who may benefit from challenging diet culture beliefs (i.e., individuals with elevated levels of ED pathology and weight stigma concerns and lower levels of fat acceptance).

The comparable levels of ED pathology between the groups may in part be due to similarities between the individuals who were reached through recruitment. Specifically, participants who interact with the research lab’s website and social media accounts may have more commonalities with one another (e.g., elevated ED pathology, weight stigma concerns), regardless of which framing they selected. Such commonalities may have attenuated possible differences between groups.

While not statistically significant, the participants in the *Diet Culture Intervention* were slightly younger and more likely to identify as multiracial than those in the *Wellness Resource*. While social media can be harmful in terms of perpetuating diet culture messages, certain content may be protective, such as anti-diet messages (e.g., content challenging messages about “unhealthy” foods or exercise solely for weight management) [36]. With increasing rates of social media use among young individuals [37] and the increase in visibility of movements such as fat activism and body positivity on these platforms [38], younger participants in this study may have been exposed to more anti-diet messages through their social media use and therefore may have been more aware of and more willing to oppose diet culture than older participants. There were also slightly more participants who identified as multiracial in the *Diet Culture Intervention* than the *Wellness Resource*. Cultural differences (e.g., different body ideals) and differing levels of acculturation may influence the level of exposure to and internalization of diet culture messages, though research in this area is mixed [39].

The source of recruitment (i.e., social media or undergraduate research portal) was associated with different levels of ED pathology, body image, and weight stigma concerns, with individuals recruited through social media having higher levels of ED pathology, poorer body image, and greater weight stigma concerns than those recruited through the undergraduate research portal. These results may be because many individuals who follow the research lab’s social media accounts are likely seeking ED resources or in search of ED treatment studies and therefore may be more likely than individuals from a general undergraduate research portal to have symptoms associated with EDs. Participants from social media were also more likely to identify as White and less likely to identify as Black compared to those from the research portal. This difference in ethnicity may be explained by the fact that most individuals seeking psychological ED treatment are White [40]. Participants recruited through social media were older and, unsurprisingly, less likely to be a student than those from the undergraduate research portal. Differences in ED pathology and demographic characteristics support the need for future research to also consider the source of recruitment when conducting DMHIs.

This study contributes to the overall understanding of DMHI implementation. Although the two framings used in this study resulted in recruitment of participants with similar characteristics, a study recruiting men for a DMHI found that different framing resulted in recruitment of individuals of different ages and levels of clinical severity [22]. Additionally, framing influences how many individuals enroll in a study or a DMHI. One study found that approximately half of participants were recruited using 2 of the 15 different advertisements, which varied in terms of text headlines and images [21]. When implementing a DMHI, an initial pilot study is advised to test whether the selected framing(s) of the DMHI result in the successful recruitment of the target population: individuals in need of accessible care.

The DMHI framing is also important for its potential role in increasing participant engagement and reducing dropouts. Current guidelines recommend participant choice in treatment in therapeutic settings in part to improve adherence to treatment [41]. Different framings for DMHIs might produce similar effects, as they may allow participants to select interventions that are of greater interest or perceived utility for them. Findings are mixed, however, in terms of whether participant choice of digital interventions and strength of preference for an intervention actually improves participant engagement and reduces dropout [42,43,44]. While outside the scope of the current study, more research is needed to determine whether DMHI framings are associated with higher levels of engagement and reduced dropout.

A notable strength of this study was the use of a widely accessible DMHI addressing diet culture. This DMHI is the first intervention to specifically target diet culture beliefs, and one of the study framings used popular lay terminology (“wellness”). While several DMHIs have been designed to treat EDs or reduce the risk of developing EDs [44,45,46,47], none of these studies have examined how framing or recruitment source (e.g., social media) may influence recruitment for such interventions. As a single-session DMHI, this intervention also had a low participant burden. Findings from this study should be interpreted within the limitations of the method. Although this study included measures of ED pathology, it did not incorporate a gold-standard measure of ED symptomology, such as the Eating Disorder Examination—Questionnaire [28] or the Eating Pathology Symptoms Inventory [48]. The measures that were used, while capturing symptoms associated with EDs (e.g., fear of weight gain), do not have established clinical cutoff scores, which is a limitation. Additionally, while both versions of the intervention framing were posted in the undergraduate research portal and on social media, it cannot be guaranteed that all participants saw both forms of the framing.

While the goal of the current study was to recruit individuals with elevated ED pathology, future research on diet culture DMHIs could incorporate a wider variety of recruitment outlets to improve the generalizability to the broader population. Future work should test other types of intervention framings, such as *Body Positivity Resource* or a more general framing description (e.g., “Intervention to find peace in your body”). Additional research should test intervention framings with other populations, including men and adolescents, to identify ways to increase enrollment in DMHIs that target diet culture, ED pathology, and related beliefs.

## 5. Conclusions

This study found that a DMHI framed as either a *Diet Culture Intervention* or *Wellness Resource* resulted in successful recruitment of individuals at risk of disordered eating. The *Wellness Resource* framing resulted in the recruitment of more individuals with lower fat acceptance compared to the *Diet Culture Intervention*. No other framing group differences were statistically significant, though post hoc analyses identified differences based on recruitment source (i.e., social media versus undergraduate research portal). Future research on DMHIs targeting diet culture should test other framings and use different populations.

## Figures and Tables

**Figure 1 ijerph-21-00024-f001:**
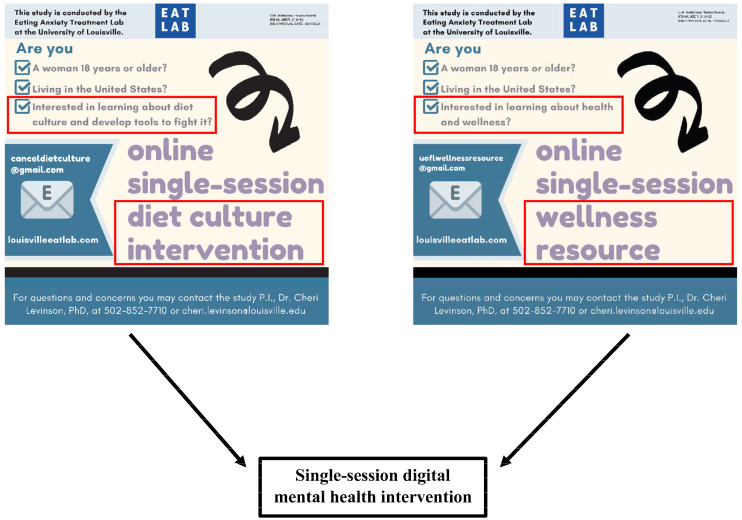
Differences in intervention framing.

**Table 1 ijerph-21-00024-t001:** Participant demographic information.

	Groups—No. (%) or Mean (SD)		
Categories	Diet Culture	Wellness Resource	Overall
	(*n* = 201)	(*n* = 196)	(*N* = 397)
Age	30.77 (13.78)	34.21 (17.24)	32.47(15.66)
Gender			
Cisgender woman	199 (99.0%)	192 (98.0%)	391 (98.5%)
Transgender woman	0 (0.0%)	0 (0.0%)	0 (0.0%)
Nonbinary	2 (1.0%)	4 (2.0%)	6 (1.5%)
Ethnicity			
American Indian or Alaskan Native	0 (0.0%)	1 (0.5%)	1 (0.4%)
Asian or Pacific Islander (includes Asian American)	17 (8.5%)	11 (5.6%)	28 (7.1%)
Black, not of Hispanic origin (includes African American)	27 (13.4%)	30 (15.3%)	57 (14.4%)
Hispanic	12 (6.0%)	17 (8.7%)	29 (7.3%)
Biracial, multiple broad categories	15 (7.5%)	4 (2.0%)	19 (4.8%)
White, not of Hispanic origin (includes Caucasian, European American, Middle Eastern)	130 (64.7%)	130 (66.3%)	260 (65.5%)
Not listed	0 (0.0%)	3 (1.5%)	3 (0.8%)
Body mass index	27.18 (8.53)	27.90 (8.94)	27.53(8.73)
Sexual orientation			
Lesbian or gay	11 (5.5%)	6 (3.1%)	17 (4.3%)
Straight/Heterosexual	145 (72.1%)	158 (80.6%)	303 (76.3%)
Bisexual	35 (17.4%)	27 (13.8%)	62 (15.6%)
Don’t know	6 (3.0%)	1 (0.5%)	7 (1.8%)
Other	2 (1.0%)	3 (1.5%)	5 (1.3%)
Prefer not to disclose	2 (1.0%)	1 (0.5%)	3 (0.8%)
Student status			
Undergraduate student	90 (44.8%)	100 (51.0%)	190 (47.9%)
Graduate student	19 (9.5%)	8 (4.1%)	27 (6.8%)
Not a student	92 (45.8%)	88 (44.9%)	180 (45.3%)

Notes. No. = number.

**Table 2 ijerph-21-00024-t002:** Measure descriptives.

	Groups—Mean (SD)		
Categories	Diet Culture	Wellness Resource	Overall
	(*n* = 201)	(*n* = 196)	(*N* = 397)
Eating Disorder Pathology			
SEDS	2.68 (1.43)	2.50 (1.41)	2.59 (1.42)
TREI	4.32 (1.59)	4.26 (1.58)	4.29 (1.58)
TREI Restriction	3.92 (1.69)	4.00 (1.61)	3.96 (1.65)
EFQ Fear of Weight Gain	5.55 (1.77)	5.33 (1.86)	5.44 (1.81)
Body Image			
TIQ Body Dissatisfaction	3.58 (0.91)	3.45 (0.93)	3.52 (0.92)
TIQ Body Ideal	3.46 (0.88)	3.48 (0.89)	3.47 (0.89)
Female Body Scale	−1.76 (1.52)	−1.75 (1.62)	−1.75 (1.57)
FFBS	−0.48 (1.81)	−0.50 (1.95)	−0.49 (1.88)
FFBS Ideal Muscularity	3.89 (1.31)	4.03 (1.49)	3.96 (1.41)
Body Appreciation Scale-2	2.91 (0.94)	2.92 (0.95)	2.92 (0.94)
Weight Stigma Concerns			
Weight Stigma Concerns Scale ^a^	4.60 (3.40)	4.20 (3.20)	4.60 (3.35)
Fat Acceptance			
Fat Acceptance Scale	4.71 (0.62)	4.47 (0.67)	4.59 (0.66)

Notes. ^a^ Median and interquartile range reported due to nonnormal distribution. SEDS = State Eating Disorder Symptom Survey; TREI = Thinness and Restricting Expectancies Inventory, EFQ = Eating Disorder Fear Questionnaire, TIQ = Thin Ideal Questionnaire.

**Table 3 ijerph-21-00024-t003:** Difference tests between diet culture intervention and wellness resource framing groups.

Instrument	*t*(ꭓ^2^)	*df*	*p*	Benjamini–Hochberg Critical *p*-Value	Cohen’s *d*
Fat Acceptance Scale	3.56	362	<0.001 *	0.003	0.37
Age	−2.20	372.61	0.029	0.006	−0.22
Ethnicity ^a^	(12.19)	--	0.038	0.009	--
Student Status	(5.04)	2	0.081	0.012	--
TIQ Body Dissatisfaction	1.31	367	0.192	0.015	0.14
Sexual Orientation ^a^	(7.05)	--	0.197	0.018	--
EFQ Fear of Weight Gain	1.19	372	0.237	0.021	0.12
SEDS	1.14	356	0.257	0.024	0.12
Weight Stigma Concerns Scale ^b^	32,611.50	--	0.268	0.026	−0.06 ^c^
FFBS Ideal Muscularity	−0.93	356	0.351	0.029	−0.10
Body Mass Index	−0.82	392	0.411	0.032	−0.08
TREI Restriction	−0.49	372	0.622	0.035	−0.05
TREI	0.35	372	0.728	0.038	0.04
TIQ Ideal Body	−0.13	367	0.895	0.041	−0.01
FFBS	0.11	356	0.912	0.044	0.01
Female Body Scale	−0.10	356	0.920	0.047	−0.01
Body Appreciation Scale-2	−0.03	372	0.977	0.050	−0.00

Notes. * statistically significant after Benjamini–Hochberg correction. ^a^ Fisher’s exact test. ^b^ Wilcoxon rank-sum test. ^c^ Effect size for Wilcoxon rank-sum test was calculated by dividing *z* statistic by square root of *N.* TIQ = Thin Ideal Questionnaire, SEDS = State Eating Disorder Symptom Survey; FFBS = Female Fit Body Scale, EFQ = Eating Disorder Fear Questionnaire, TREI = Thinness and Restricting Expectancies Inventory.

**Table 4 ijerph-21-00024-t004:** Difference tests between social media and undergraduate research portal recruitment.

Instrument	*t*(ꭓ^2^)	*df*	*p*	Benjamini–Hochberg Critical *p*-Value	Cohen’s *d*
Student Status	(260.36)	2	<0.001 *	0.003	--
Age	15.68	388.97	<0.001 *	0.006	1.48
Female Body Scale	−5.93	356	<0.001 *	0.009	−0.64
TREI Restriction	5.26	372	<0.001 *	0.012	0.55
Body Mass Index	5.22	371.84	<0.001 *	0.015	0.49
FFBS	−5.40	352.02	<0.001 *	0.018	−0.55
TIQ Body Dissatisfaction	4.94	367	<0.001 *	0.021	0.52
EFQ Fear of Weight Gain	4.96	276.37	<0.001 *	0.024	0.54
Weight Stigma Concerns Scale ^b^	24,006.00	--	<0.001 *	0.026	−0.25 ^c^
TREI	4.79	298.33	<0.001 *	0.029	0.52
SEDS	3.89	283.32	<0.001 *	0.032	0.43
Body Appreciation Scale-2	−3.30	372	0.001 *	0.035	−0.35
Ethnicity ^a^	(17.97)	--	0.003 *	0.038	--
TIQ Ideal Body	2.94	367	0.004 *	0.041	0.31
FFBS Ideal Muscularity	1.07	356	0.286	0.044	0.12
Sexual Orientation ^a^	(4.49)	--	0.488	0.047	--
Fat Acceptance Scale	−0.50	362	0.618	0.050	−0.05

Notes. * statistically significant after Benjamini–Hochberg correction. ^a^ Fisher’s exact test. ^b^ Wilcoxon rank-sum test. ^c^ Effect size for Wilcoxon rank-sum test was calculated by dividing *z* statistic by square root of *N.* TIQ = Thin Ideal Questionnaire, SEDS = State Eating Disorder Symptom Survey; FFBS = Female Fit Body Scale, EFQ = Eating Disorder Fear Questionnaire, TREI = Thinness and Restricting Expectancies Inventory.

## Data Availability

The data that support the findings of this study are available from the corresponding author upon reasonable request.

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
