# Peer review of "Digital Mental Health Interventions: Differences in Diet Culture Intervention Framing"

_ijerph, 2023, doi:10.3390/ijerph21010024_

Round 1
Reviewer 1 Report
Comments and Suggestions for Authors

Reviewer 2 Report
Comments and Suggestions for Authors
Good morning.
I would like to congratulate the authors for their efforts in their work. They present interesting data. However, I consider that for the improvement of the final paper and its acceptance, a number of changes should be made.
Firstly, the type of manuscript is Brief Report and appears as Article. I believe that the paper provides information that could be expanded to be a full article. But that is the decision of the authors and the editor.
The main improvements should be made in the Materials and Methods section. The text in the proposed order complicates the comprehension and follow-up of the article, as in order to understand some of the data it is necessary to move forward or backward in the pages. The order should be:
Materials and Methods
2.1. Participants
2.2. Measures
2.3. Procedures
2.4. Data Analyses (remove from results section)
Table 1 should only contain the Participant Demographic Information, while the descriptions of the questionnaires should appear under Results. Therefore, I recommend splitting the table into two parts.
The discussion should be expanded, as it is somewhat limited. The results should be interpreted in relation to other studies that have looked at the same variables or related topics.
The important limitations of the study should also be pointed out, not just the strengths.
